# An Examination of Fungal and Bacterial Assemblages in Bulk and Rhizosphere Soils under *Solanum tuberosum* in Southeastern Wyoming, USA

**Gordon F. Custer** [1,2,3,*], **Linda T. A. van Diepen** [1,2] **and William Stump** [3]

[1] Program in Ecology, University of Wyoming, Laramie, WY 82072, USA
[2] Department of Ecosystem Science and Management, University of Wyoming, Laramie, WY 82072, USA
[3] Department of Plant Sciences, University of Wyoming, Laramie, WY 82072, USA
[*] Correspondence: gordon.custer91@gmail.com

**Abstract:** *Solanum tuberosum*, commonly known as potato, is the most important non-cereal crop in the world. However, its cultivation is prone to disease and other issues. In recent years, a newfound interest in the soil microbiome and the potential benefits it may convey has led researchers to study plant–microbe interactions in great detail and has led to the identification of putative beneficial microbial taxa. In this survey, we examined fungal and bacterial diversity using high-throughput sequencing in soils under a potato crop in southeastern Wyoming, USA. Our results show decreased microbial diversity in the rhizosphere, with increases in the abundances of arbuscular mycorrhizal fungi as well as pathogenic microbes. We show coarse taxonomic differences in microbial assemblages when comparing the bulk and rhizosphere soils for bacteria but not for fungi, suggesting that the two kingdoms respond differently to the selective pressures of the rhizosphere. Using cooccurrence network analysis, we identify microbes that may serve as keystone taxa and provide benefits to their host plants through competitive exclusion of detrimental pathogenic taxa and increased nutrient availability. Our results provide additional information on the structure and complexity of the potato rhizosphere microbiome and highlight candidate taxa for microbial isolation and inoculation.

**Keywords:** ITS; 16S; microbial ecology; agroecology; community





## 1. Introduction

Rapidly increasing human population coupled with predicted climate change scenarios have the potential to manifest in food supply vulnerability [1]. In order to provide a stable food supply in the face of changing climate and other disturbances, researchers and farmers alike are turning to microorganisms, specifically the soil microbiome. Soil microorganisms are known to dramatically affect ecosystem properties such as nutrient cycling [2,3] and ecosystem productivity [4]. Of particular importance, the microbial assemblage directly surrounding plant roots, known as the rhizosphere microbiome, plays a large role in plant fitness via mediating abiotic stress [5,6], increasing plant nutrient access [2,7–9], and affecting susceptibly to both herbivory [10,11] and pathogens [12]. There is growing evidence that plants are active participants in the recruitment of soil microbes [13,14], through the production of root exudates [15]. Together, the rhizosphere assemblage along with the host plant can be thought of as an extended phenotype upon which natural selection is able to act [16,17]. Thus, rhizosphere assemblages that increase host fitness are positively selected for, and those that act as a detriment to their host are selected against.

As the most important non-cereal crop in production agriculture, potatoes (*Solanum tuberosum*) are critical to the global food supply [18]. However, many plant pathogens, both bacterial and fungal, are known to detrimentally affect production [19]. In fact, the well-known Irish potato famine, that led to more than one million Irish citizens immigrating to the United States, was caused by none other than the devastating oomycete, *Phytophthora*

*infestans* [20]. Other pathogenic microbes of potato include *Streptomyces* spp. (Common scab), *Fusarium* spp. (Fusarium dry rot), and *Alternaria solani* (Early blight) which, similar to *Phytophtora*, can negatively affect crop yield. As the soil microbiome has been demonstrated to convey resistance to pathogens [12,21,22], an understanding of microbial populations associated with valuable crop species is incredibly important. In particular, knowing which microbes are selected by host plants to be members of the rhizosphere assemblage can provide many avenues to boost production, including the development of synthetic communities which many be used to deterministically alter the function and composition of the soil microbiome [23].

In order to better understand the composition of the potato rhizosphere microbiome, we conducted a survey of both the bulk and rhizosphere microbial assemblages in a potato monoculture in southeastern Wyoming, USA. As both bacteria and fungi are important members of the soil microbiome, we examined both Kingdoms using high-throughput sequencing of the 16s rRNA and fungal internal transcribed spacer (ITS) genes, respectively. Other studies examining the potato rhizosphere have not reported bacteria and fungi in combination [24,25]. In this short communication, the soil microbiome of *Solanum tuberosum* cv. Atlantic is examined via paired bulk and rhizosphere soil samples. We hypothesized that the rhizosphere of *Solanum tuberosum* cv. Atlantic would selectively exclude pathogenic bacteria and fungi while increasing the relative abundance of beneficial microbes like plant growth promoting bacteria and arbuscular mycorrhizal fungi (AMF). In addition, we expected alpha diversity metrics to be lower in the rhizosphere compared to the bulk soil, with strongly differentiated assemblages as determined by β-diversity.

## 2. Materials and Methods

In June of 2016, soil samples were collected from the University of Wyoming's Sustainable Agriculture Research and Extension (SAREC) facility located in Lingle, WY (42.129007142840706, −104.39168216388198). The soil at this site has an alkaline pH (~8) with $CaCO_3$ content between 1% and 3%. The soil can be characterized as silty clay loam [26]. Soil samples were collected from a field planted with *Solanum tuberosum* cv. Atlantic while plants were at the tuber bulking stage. Additional sample collections were planned, but due to a large hailstorm, the entire crop was lost. This limited our sampling to a single time point. Focal plants were randomly selected from a potato field ~0.5 ha in size. Selected focal plants were no closer than 3 m to each other and at least 3 m from the edge of the field. Soil cores were taken at 15 cm from the main plant stem. A total of nine soil samples were collected and transported on ice back to the University of Wyoming's soil microbial ecology lab.

Once back in the lab, samples were kept at 4 °C until processing, which occurred within 48 h. First, roots were collected from the soil sample with ethanol flame-sterilized forceps and placed in sterile 50 mL tubes. The remaining soil was passed through a 2 mm sieve, and was considered bulk soil. Collected roots were dry vortexed for approximately 90 s in order to remove tightly adhering soil [27,28]. A final cleaning step of removing root fragments was performed, and we defined this fraction of soil as the rhizosphere. Subsamples of the bulk and rhizosphere soils (~250 mg) were added to individual MoBio power soil tubes (MO BIO, Carlsbad, CA, USA) and frozen at −20 °C. The two soils discussed above (rhizosphere and bulk) are referred to as "soil origins" from here on.

Following sample processing, fresh soil was used for the analysis of gravimetric water content by oven drying at 105 °C for 24 h. Electrical conductivity (EC) and pH were measured using an Oakton PC700 (Oakton Instruments, Vernon Hills, IL, USA) with a soil to DI water ratio of 1:2 (*w/v*). These analyses were completed only for the bulk samples as the rhizosphere samples did not have sufficient material.

### 2.1. DNA Extraction and Library Preparation

Frozen bead tubes containing soil subsamples were thawed at room temperature, and DNA was extracted according to manufacturer instructions (MO BIO, Carlsbad, CA, USA).

DNA extracts were then frozen at −20 °C until amplification. Briefly, bacterial (16S rRNA) and fungal (internal transcribed spacer, ITS) amplicon libraries were prepared using 8-bp molecular identification indices on both the forward and reverse primers. Polymerase chain reaction (PCR) of the V4 region of the 16S rRNA gene of the bacterial genomes was done using the modified 515F (5′-GTGYCAGCMGCCGCGGTAA-3′) [29] and 806R (5′-GGACTACNVGGGTWTCTAAT-3′) [30] primers and the following conditions: 98 °C for 30 s, 30 cycles of 98 °C for 10 s, 65 °C for 10 s, 72 °C for 8 s, and 72 °C for 5 min. The ITS2 region of the fungal genomes was amplified using the fITS7 (5′-GTGARTCATCGAATCTTTG-3′) [31] and the ITS4 (5′-TCCTCCGCTTATTGATATGC-3′) [32] primers. The conditions for this amplification were as follows: 98 °C for 30 s, 30 cycles of 98 °C for 10 s, 60 °C for 10 s, 72 °C for 8 s, and 72 °C for 5 min. Positive and negative controls were included in each round of PCR. Each of the triplicate PCR reactions consisted of 0.2 µL of Phusion high-fidelity DNA polymerase, 4 µL of 5X Phusion Green HF buffer, 0.4 µL of deoxynucleotide triphosphates (10 µM), 12.4 µL of diethyl pyrocarbonate-water, 1 µL each of forward and reverse primer (10 µM), and 1 µL of template DNA. PCR products were verified on 1.5% agarose gel, and successful reactions were combined in triplicate. Products were cleaned using Axygen's AxyPrep Mag PCR Clean-up Kit according to manufacturer instructions (Axygen Biosciences, Union City, CA, USA). Concentrations of the cleaned, amplified DNA were measured using a dsDNA HS assay kit on a Qubit 3.0 fluorometer (Invitrogen/Life Technologies, Carlsbad, CA, USA). Bacterial and fungal samples were combined at equimolar concentrations in separate libraries. Both libraries were sent to the University of Minnesota's Genomics Center (UMGC) for sequencing on the Illumina MiSeq platform with V2 chemistry (2 × 250 bp). Our final libraries did not include positive or negative controls for sequencing.

*2.2. Sequence Data Analysis*

Sequence reads were analyzed from raw reads in R (Version 3.6.1) [33] using the DADA2 pipeline (Version 1.16) [34]. Filtering was performed slightly differently for ITS and 16S rRNA data due to the inherent nature of ITS reads being variable lengths. ITS filtering used the following parameters: filterAndTrim(maxEE = (1,1), truncQ = 11, maxN = 0, minLen = 50, rm.phix = TRUE). Filtering of 16S rRNA reads used: filterAndTrim (truncLen = c(240,160), maxEE = (2,2), truncQ = 10, maxN = 0, rm.phix = TRUE). One million reads were used to learn errors for both the ITS and 16S rRNA data. Chimeras were removed using DADA2's function removeBimeraDenovo, and taxonomy was assigned using taxonomic reference databases (Silvia V138 [35] for bacterial assignments and UNITE [36] for fungal assignments). Processed reads were then transferred to the Phyloseq package for statistical analyses and visualization [37].

*2.3. Statistical Analysis of Sequence Data*

We first removed any reads that were not assigned to fungi or bacteria. The remaining dataset was then transformed to within site proportional abundances to avoid rarefaction (reads assigned to an ASV/total reads within that sample). The only analysis which required rarefaction was alpha diversity measurements. All other analyses use and report proportional abundances or un-normalized data. Shannon diversity (H′) and species richness were estimated using the Phyloseq function estimate_richness, and statistical differences were determined using a one-way ANOVA. In situations where the assumptions of ANOVA could not be met using log-transformations, the non-parametric alternative Kruskal–Wallis test was utilized. This test was used for the majority of comparisons due to the non-normality of residuals. β-diversity of both bacterial and fungal samples were analyzed separately using Bray–Curtis dissimilarities. β-diversity was visualized with PCoA using the ape package [38], and significant differences between soil origins were determined via ADONIS testing using the vegan package [39].

The DESeq2 package [40] was used to test for differential abundances of bacterial and fungal taxa between soil origins. Differentially abundant taxa for both the genus

and phylum level are reported at $\alpha = 0.01$ for bacteria and $\alpha = 0.05$ for fungi. Different alpha levels were chosen due to the number of potential comparisons and indicator taxa. Functional guilds and trophic modes were assigned to fungal taxa using FUNGuild [41]. Significant differences in the proportional abundances of trophic modes and fungal guilds were determined using Kruskal–Wallis testing due to the inability to meet the assumptions of ANVOA. FUNGuild was used to assign guild and trophic mode information to fungal indicator taxa as determined by DESeq2.

Finally, cooccurrence networks were constructed using the backbone package [42]. Cooccurrence networks are commonly utilized in microbial ecology to identify taxa that are important for microbiome structure. We constructed cooccurrence networks for bacteria and fungi independently. Briefly, the bipartite graph contained 668 agents (taxa) and 16 artifacts (samples) for the fungal network and 4325 agents (taxa) and 15 artifacts (samples) for the bacterial network. From this we obtained the weighted bipartite projection and extracted its signed backbone. Edges were retained if their weights were statistically significant ($\alpha = 0.001$) by comparison to a null hypergeometric model [43]. Significant nodes (ASVs) were determined by keeping only those with at least one significant edge.

## 3. Results

### 3.1. Fungal Diversity

Fungal samples had an average of 134,599 reads per sample prior to processing. After quality filtering and chimera removal, we retained an average of 86,283 reads per sample (64.1% retention). The remaining reads were clustered into a total of 683 amplicon sequence variants (ASVs). When necessary, the ASV table was rarefied to 39,646 reads per sample, and after rarefaction 668 ASVs remained. We report nine independent bulk and seven independent rhizosphere samples for our statistical analyses.

Analysis of $\alpha$-diversity metrics showed significant differences between bulk and rhizosphere soils for fungal richness ($p < 0.05$) but not Shannon diversity ($p = 0.36$), with rhizosphere soils having lower species richness (Table 1). Assessment of fungal $\beta$-diversity with PERMANOVA testing revealed soil origin ($p < 0.01$, $F_{1,11} = 3.221$, $R^2 = 0.172$) and soil moisture content ($p < 0.01$, $F_{1,11} = 2.498$, $R^2 = 0.133$) to be significant predictors (Figure 1).

**Table 1.** Alpha diversity metrics and summary statistics for bacteria and fungi and soil origins.

| | | **Fungi** | | | | |
| --- | --- | --- | --- | --- | --- | --- |
| | | | **Bulk** | | **Rhizosphere** | |
| | | | Mean | Standard Deviation | Mean | Standard Deviation |
| Shannon Diversity (H′) | *p*-value | $p = 0.36$ | 3.54 | 0.28 | 3.67 | 0.09 |
| | Test Statistic | $X^2_{1,14} = 0.891$ | | | | |
| Richness | *p*-value | $p < 0.05$ | 153.33 | 21.99 | 130.29 | 19.99 |
| | Test Statistic | $F_{1,14} = 4.672$ | | | | |
| | | **Bacteria** | | | | |
| | | | **Bulk** | | **Rhizosphere** | |
| | | | Mean | Standard Deviation | Mean | Standard Deviation |
| Shannon Diversity (H′) | *p*-value | $p < 0.0001$ | 6.42 | 0.26 | 5.75 | 0.28 |
| | Test Statistic | $F_{1,14} = 22.29$ | | | | |
| Richness | *p*-value | $p = 0.118$ | 1227 | 384 | 952 | 218 |
| | Test Statistic | $F_{1,14} = 2.796$ | | | | |

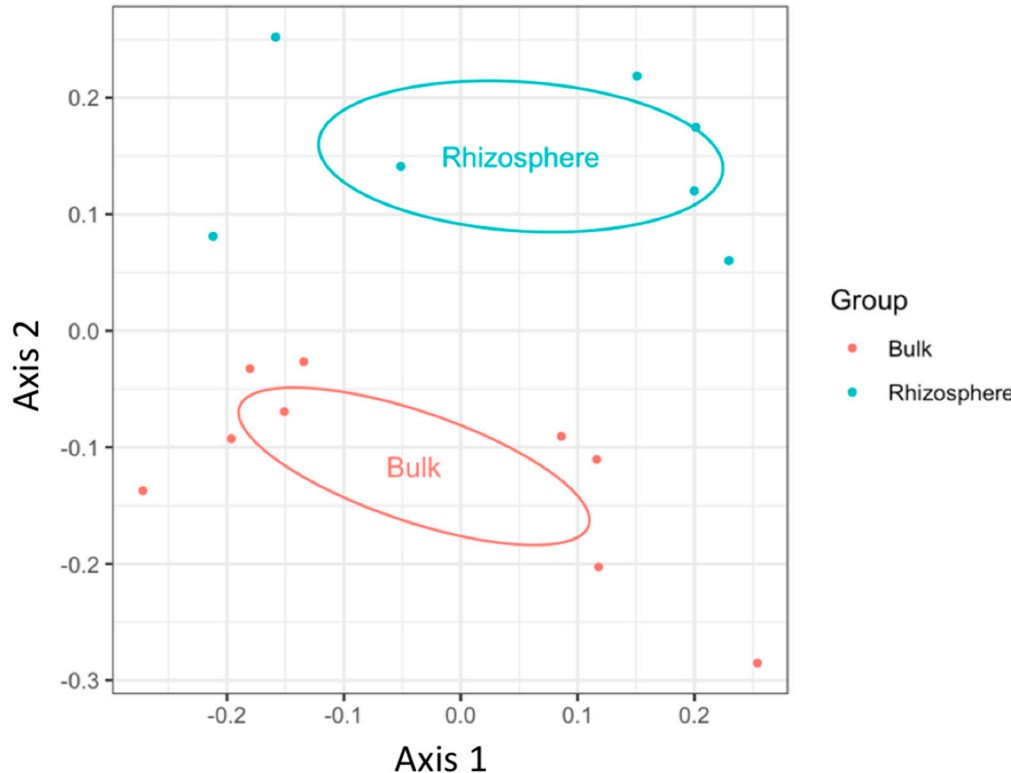

**Figure 1.** Principal coordinate analysis (PCoA) of fungal assemblage based upon Bray-Curtis dissimilarity. Soil origin and moisture are significant predictors of fungal β-diversity ($p < 0.01$). Points indicate individual samples, and ellipses represent the 95% confidence interval of the mean of each soil origin.

FUNGuild assigned 439 of the 683 total ASVs. Of those 439 ASVs, 313 were assigned with highly probable or probable confidence. Only assignments with highly probable or probable confidence were used for downstream analysis. The trophic modes saprotroph ($p < 0.01$) and pathotroph-saprotroph-symbiotroph ($p < 0.01$) were elevated in the bulk soil samples, while pathotroph-saprotroph ($p < 0.01$) was elevated in the rhizosphere samples (Table 2). The fungal guilds AMF ($p < 0.01$), dung-saprotroph-soil-saprotroph-wood-saprotroph ($p < 0.05$), fungal-parasite-plant-pathogen-plant-saprotroph ($p < 0.001$), and plant-pathogen-plant-saprotroph ($p < 0.05$) were elevated in the rhizosphere samples, while the dung-saprotroph-undefined-saprotroph ($p < 0.05$) and undefined-saprotroph ($p < 0.01$) were elevated in the bulk samples

Differential abundance testing with the DESeq2 package showed twelve genus-level taxa to be significantly different among the bulk and rhizosphere soil samples (Figure 2). Of the 12 indicator taxa, seven were indicative of bulk soil and five of rhizosphere samples. Eleven were members of Ascomycota, and only one was a member of the phylum Basidiomycota (Supplementary Table S1). Guild and trophic mode assignments show that saprotrophs were the dominant guild of indicators for bulk soil samples, with five of the seven being assigned as such (*Ophiosphaerella*, *Ascobolaceae*, *Kotlabaea*, *Pseudaleuria*, and *Botryotrichum*). Only one of the two non-saprobe indicators, *Dendryphion*, was assigned by FUNGuild as a plant pathogen and pathotroph (Supplementary Table S1). As for rhizosphere soil indicators, *Ilyonectria* was assigned as a plant pathogen and pathotroph. The other four genus-level indicators were assigned as both plant pathogens and saprotrophs (Supplementary Table S1). No fungal phyla were differentially abundant between the bulk and rhizosphere samples. (Table 2).

*Fusarium* was the most common genus in 15 of the 16 samples, accounting for on average ~18% of the total fungal reads for each sample. Only one bulk sample had *Conocybe*

as the most dominant, though *Fusarium* was the second most common in that sample. The most common *Fusarium* ASVs, as per BLASTn assignment, included several sequence variants of *Fusarium equiseti*, *Fusarium solani*, and *Fusarium oxysporoum* (*Fusarium* Phyloseq object containing all ASVs and taxonomy can be found in our Supplementary Materials).

Cooccurrence network analysis revealed six fungal ASVs to be significant at $\alpha = 0.001$. Of those taxa, two were unassigned past the kingdom level, and BLASTn assignment shows them to be likely originating from the potato genome, though they had fungal taxonomy assigned via DADA2. The two unassigned ASVs were dominant in the rhizosphere samples. Of the ASVs that were assigned taxonomy past the phylum level, the most common genera included *Fusairum* (two assignments). These sequence variants were more abundant in the rhizosphere samples. The other two genera included *Verticillium* and *Rhizophylctis*. It is worth noting that each node only contained a single significant edge, and therefore may not be very useful for determining the importance and ecology of fungal ASVs in our survey.

**Table 2.** Summary statistics of FUNGuild assignments for trophic mode and guild by soil origin. Relative abundance means are reported as percent abundance within a single sample, i.e., within sample proportional abundance.

| | | | Bulk | | Rhizosphere | |
|---|---|---|---|---|---|---|
| **Fungal Trophic Modes** | | | | | | |
| | | | Mean | Standard Deviation | Mean | Standard Deviation |
| Saprotroph | *p*-value | $p < 0.01$ | 0.1861% | 0.0337% | 0.1170% | 0.0283% |
| | Test Statistic | $X^2_{1,14} = 8.371$ | | | | |
| Pathotroph-Saprotroph | *p*-value | $p < 0.01$ | 0.0770% | 0.0170% | 0.1159% | 0.0187% |
| | Test Statistic | $X^2_{1,14} = 7.085$ | | | | |
| Pathotroph-Saprotroph-Symbiotroph | *p*-value | $p < 0.01$ | 0.0097% | 0.0086% | 0.0024% | 0.0030% |
| | Test Statistic | $X^2_{1,14} = 4.371$ | | | | |
| **Fungal Guilds** | | | | | | |
| | | | Bulk | | Rhizosphere | |
| | | | Mean | Standard Deviation | Mean | Standard Deviation |
| Arbuscular-Mycorrhizal | *p*-value | $p < 0.01$ | 0.0021% | 0.0037% | 0.0087% | 0.0075% |
| | Test Statistic | $X^2_{1,14} = 4.389$ | | | | |
| Dung-Saprotroph-Soil-Saprotroph-Wood-Saprotroph | *p*-value | $p < 0.05$ | 0.0154% | 0.0170% | 0.0169% | 0.0448% |
| | Test Statistic | $X^2_{1,14} = 5.980$ | | | | |
| Dung-Saprotroph-Undefined-Saprotroph | *p*-value | $p < 0.05$ | 0.1314% | 0.1572% | 0.0389% | 0.0188% |
| | Test Statistic | $X^2_{1,14} = 6.1875$ | | | | |
| Fungal-Parasite-Plant-Pathogen-Plant-Saprotroph | *p*-value | $p < 0.001$ | 0.0227% | 0.0064% | 0.0882% | 0.0611% |
| | Test Statistic | $X^2_{1,14} = 11.117$ | | | | |
| Plant-Pathogen-Plant-Saprotroph | *p*-value | $p < 0.05$ | 0.0010% | 0.0030% | 0.0076% | 0.0077% |
| | Test Statistic | $X^2_{1,14} = 4.509$ | | | | |
| Undefined-Saprotroph | *p*-value | $p < 0.01$ | 0.1876% | 0.0375% | 0.1156% | 0.0267% |
| | Test Statistic | $X^2_{1,14} = 9.100$ | | | | |

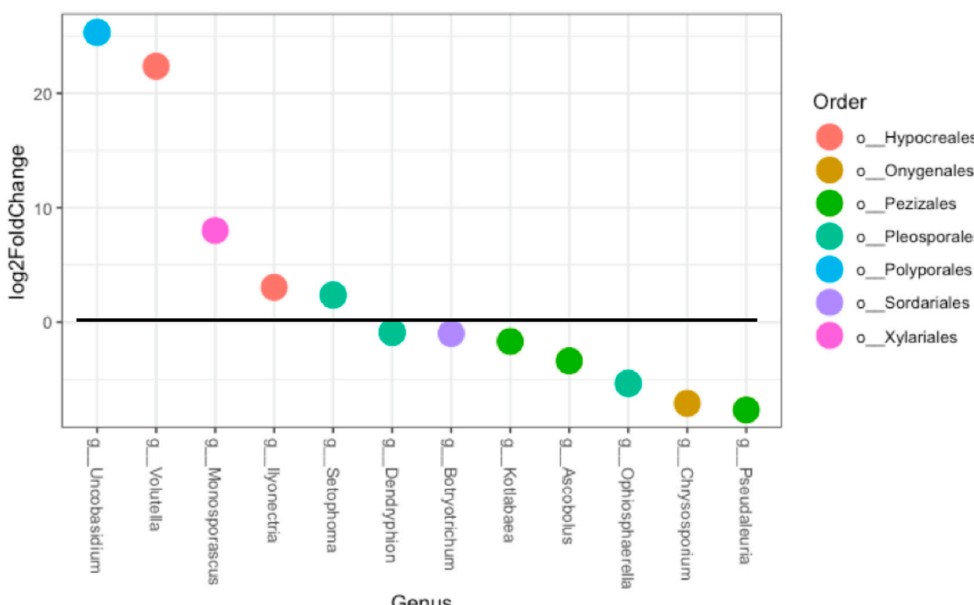

**Figure 2.** Visualization of differentially abundant fungal genera ($\alpha$ = 0.05). Each point represents an ASV that was identified by DESeq2 as differentially abundant between soil origins. Points are colored by order-level taxonomy. The *y*-axis indicates log$_2$-fold change in abundance. A positive log$_2$-fold change indicates that ASV is associated with the rhizosphere samples. A negative log$_2$-fold change indicates that ASV is associated with the bulk soil samples.

### 3.2. Bacterial Diversity

Bacterial samples had an average of 58,948 reads prior to processing, and after quality filtering and chimera removal, an average of 44,117 reads per sample remained (74.8% retention). The remaining reads were clustered into a total of 4891 ASVs. When necessary, the ASV table was rarefied to 12,832 reads per sample. After rarefaction, 4325 ASVs remained. We report eight independent bulk and seven independent rhizosphere bacterial samples for our statistical analyses, as one sample failed to sequence.

Analysis of $\alpha$-diversity metrics showed significantly lower diversity in the rhizosphere compared to the bulk soil for bacterial Shannon diversity (H') ($p < 0.0001$) but not for bacterial richness ($p = 0.118$) (Table 1). PERMANOVA testing revealed only soil origin ($p < 0.0001$, $F_{1,13} = 12.409$, $R^2 = 0.4884$) to be a significant predictor of bacterial $\beta$-diversity (Figure 3).

Differential abundance testing with the DESeq2 package revealed a total of 65 genus-level taxa to be differentially abundant in either bulk or rhizosphere soils (visualized at the family level Figure 4a, and phylum level 4b, Supplementary Table S2). Of the 65 genera, 41 were indicative of rhizosphere soil samples and 24 of bulk soil samples. All but one of the assigned indictors of Actinobacteria and Proteobacteria were associated with the rhizosphere samples (Figure 4b). The Chloroflexi indicator was associated with the rhizosphere samples. The members of Bacteroidota, Verrucomicrobiota and Acidobacteriota were split between the two soil origins. Indicators assigned to the phyla Plancomycetota, Nitrospirota, Gemmatimonadota, Abditbacteriota, and Fibrobacterota were solely associated with bulk soil samples (Supplementary Table S2). Indicator analyses at the level of phylum showed eight phyla to be differentially abundant between the rhizosphere or bulk samples (Table 3, Figure 4b). Five phyla including Cyanobacteria, Actinobacteria, Chloroflexi, Proteobacteria and Myxococcota were associated with the rhizosphere samples, while Gemmatimonadota, Armatimonadota and Abitibacteriota were associated with the bulk samples. Bar plots of phylum level abundances show strong differentiation between bulk and rhizosphere bacterial assemblages (Figure 5) even at coarse taxonomic resolution. Bulk samples also showed higher proportions of Acidobacteriota and Verrucomicrobiota, though this difference was not significant.

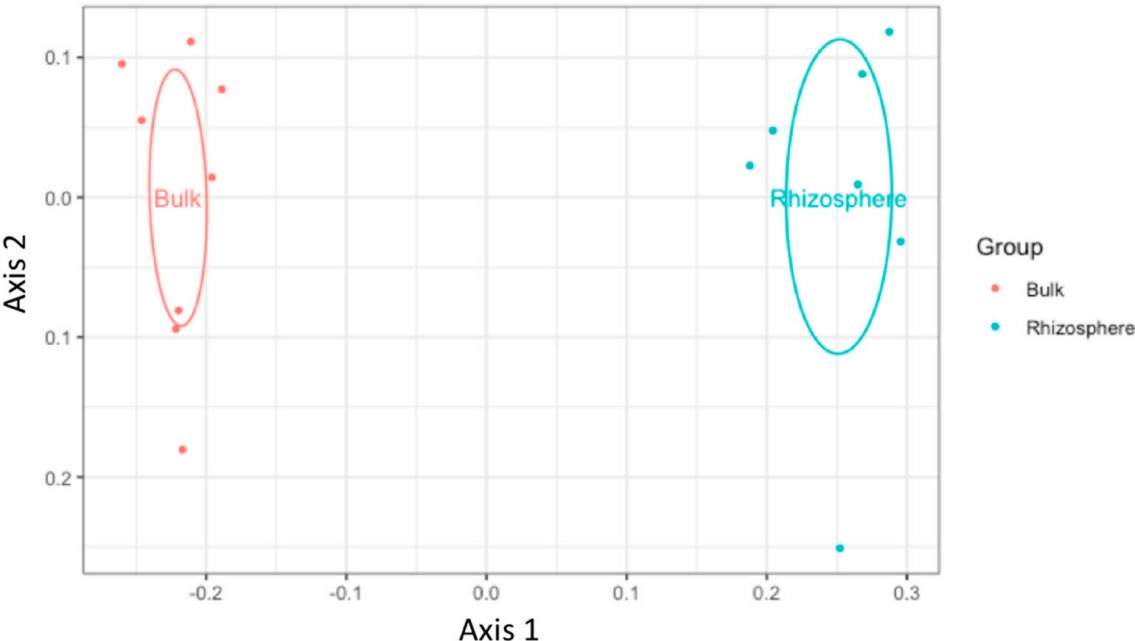

**Figure 3.** Principal coordinate analysis (PCoA) of bacterial assemblage based upon Bray-Curtis dissimilarity. Soil origin was the lone significant predictor of bacterial β-diversity ($p < 0.0001$). Points indicate individual samples, and ellipses represent the 95% confidence interval of the mean of each soil origin.

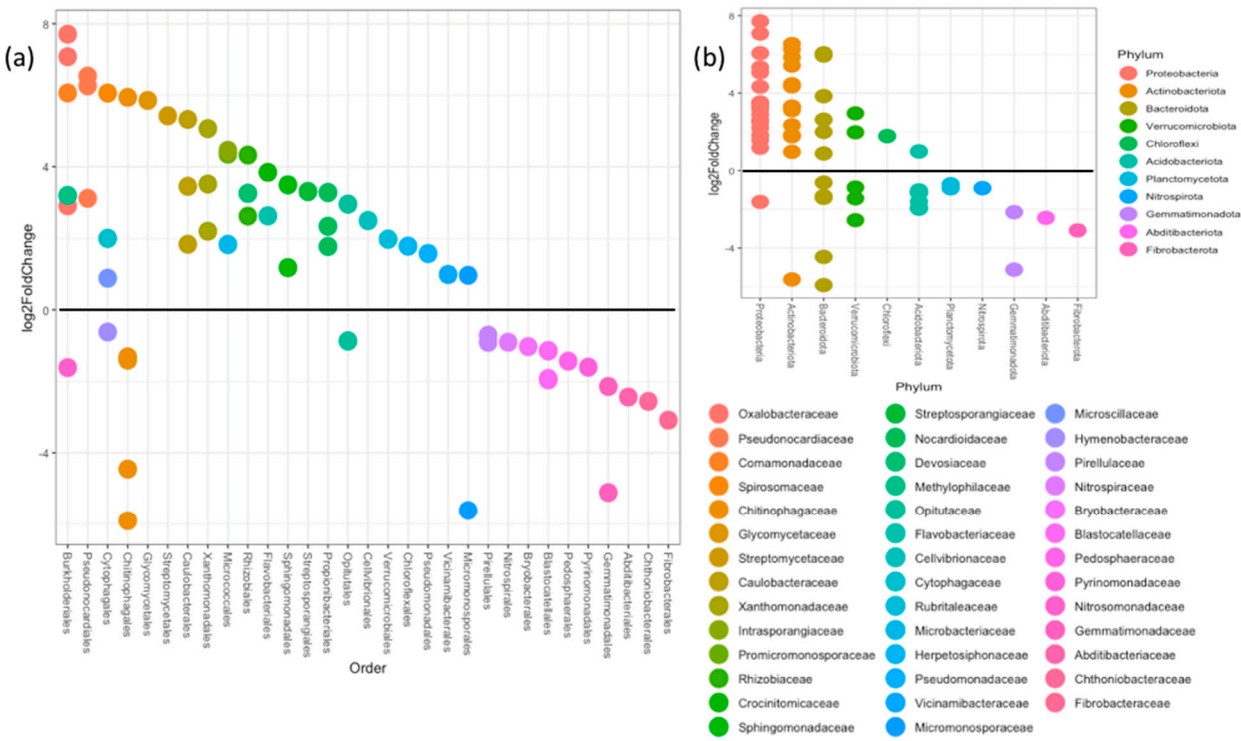

**Figure 4.** (**a**) Visualization of differentially abundant bacterial genera. Each point represents an ASV that was identified by DESeq2 as differentially abundant between soil origins. Points are colored based on their family level taxonomy. The *y*-axis indicates $\log_2$-fold change in abundance. A positive $\log_2$-fold change indicates that ASV is associated with the rhizosphere samples. A negative $\log_2$-fold change indicates that ASV is associated with the bulk soil samples. (**b**) Visualization of differentially abundant bacterial genera visualized at the phylum level ($\alpha = 0.01$).

**Table 3.** Differentially abundant bacterial taxa at the phylum level (*p* < 0.01) as per DESeq2. Mean rarefied abundance represents the mean abundance across all soil samples. Change (Log$_2$ fold) indicates the multiplicative change in taxon abundance between soil origins. Negative numbers indicate an association with bulk samples, and positive numbers indicate an association with rhizosphere samples. The adjusted *p*-value column show the FDR corrected *p*-value.

| Mean Relative Abundance | Change (log$_2$ Fold) | Association | Adjusted *p*-Value | Kingdom | Phylum |
|---|---|---|---|---|---|
| 2780.41682 | 6.905558924 | Rhizosphere | $7.25 \times 10^{-164}$ | Bacteria | Cyanobacteria |
| 5773.55747 | 2.114174238 | Rhizosphere | $7.78 \times 10^{-28}$ | Bacteria | Actinobacteriota |
| 3572.14083 | 0.565324082 | Rhizosphere | $1.52 \times 10^{-5}$ | Bacteria | Chloroflexi |
| 6986.19763 | 1.230783697 | Rhizosphere | $3.50 \times 10^{-19}$ | Bacteria | Proteobacteria |
| 851.536029 | 0.92383889 | Rhizosphere | 0.0005466 | Bacteria | Myxococcota |
| 764.000925 | −0.727541596 | Bulk | $1.12 \times 10^{-7}$ | Bacteria | Gemmatimonadota |
| 218.042158 | −1.157547177 | Bulk | $1.23 \times 10^{-8}$ | Bacteria | Armatimonadota |
| 14.6936551 | −1.880961129 | Bulk | 0.00190362 | Bacteria | Abditibacteriota |

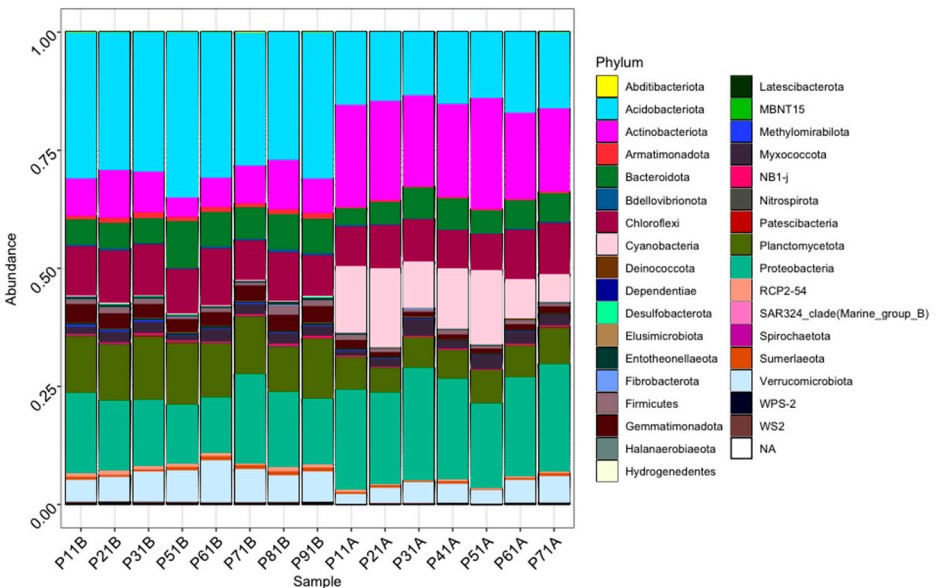

**Figure 5.** Bar plots of within sample proportional abundances of bacteria at the phylum level with samples arranged by soil origins. Samples ending in "A" indicate rhizosphere soil samples. Samples ending in "B" indicate bulk soil samples.

While the most abundant genus of fungi was the same across both soil origins, there was a clear divide in the most abundant bacterial genus based on soil origin. The genus *Streptomyces* was the most abundant in all seven rhizosphere samples, accounting for an average of ~4.3% of sequence reads. In the bulk samples, *RB41* was the most abundant in eight of the nine bulk samples and accounted for ~5.9% of the total reads on average. While *RB41* was also common in the rhizosphere, being one of the top ten genera in six of the seven samples and accounting for an average ~1.7% of the total reads, *Streptomyces* was not in the top ten most abundant genera for any of the bulk samples. It accounted for an average of only 0.01% of the total reads for bulk samples. There were 12 different *Streptomyces* ASVs found in the bacterial dataset. The most abundant *Streptomyces* ASV was assigned to the pathogen *Streptomyces scabiei* by the BLASTn database. We also report an immediate drop in the relative abundance of *Streptomyces* taxa after the most abundant, dropping from a mean relative abundance of ~3.5% to 0.5%.

Cooccurrence network analysis revealed 175 bacterial ASVs to have significant edges at α = 0.001. Comparison of the relative abundance of only those ASVs determined to be significant nodes showed differentiation between bulk and rhizosphere samples (Figure 6). Bulk samples had higher proportions of Planctomycetota, Gemmatimonadota,

and Acidobactiota, while rhizosphere samples contained higher relative abundances of Actinobacteriota, Cyanobacteria, and Proteobacteria. There were 20 bacterial ASVs (nodes) that had seven or more significant edges. They represent the phyla Planctomycetota (3), Verrucomicrobiota (1), Acidobacteriota (2), Actinobacteriota (3), Cyanobacteria (5), and Proteobacteria (3) among others. The most abundant genera with significant node taxa were *RB41* and *Pirellula* in bulk soils, with *Massilia*, *Amycolatpsis*, *Streptomyces*, and *Lechevaleria* being abundant in the rhizosphere samples.

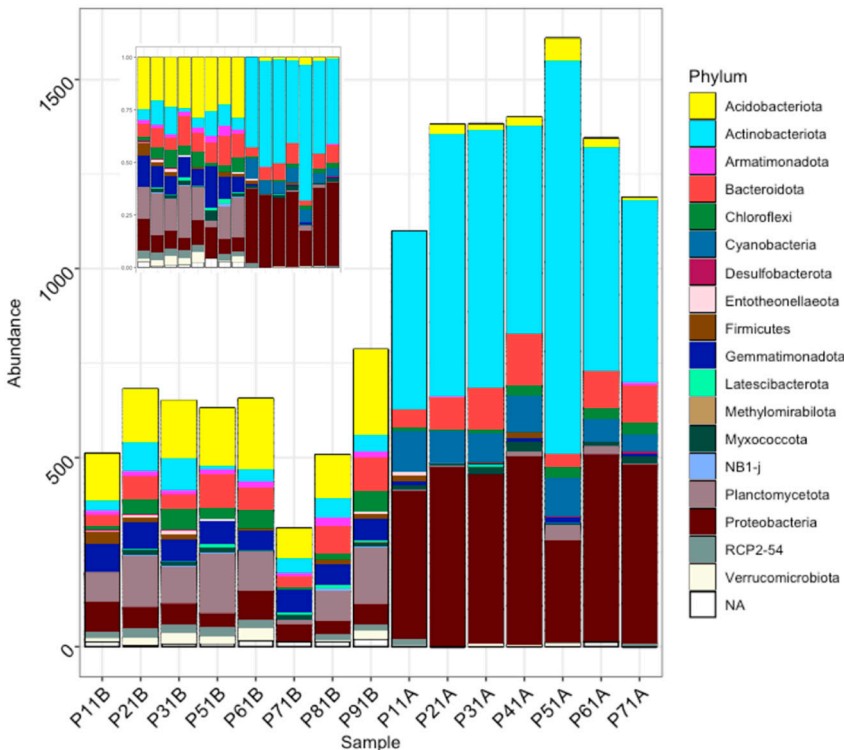

**Figure 6.** Bar plots of rarefied read counts and within sample proportional abundances (insert) of significant nodes as determined by our cooccurrence networks colored at the phylum level. Samples ending in "A" indicate rhizosphere soil samples. Samples ending in "B" indicate bulk soil samples.

## 4. Discussion

Our analyses show that differences in α-diversity metrics between bulk and rhizosphere soils were specific to bacteria or fungi, and that in most cases, the rhizosphere displayed lower α-diversity. In support of our initial hypotheses, β-diversity revealed significant differences between soil origins for both bacteria and fungi, in accordance with the existing literature on potato microbiomes [44]. However, soil moisture content was only a significant predictor of β-diversity for fungi, suggesting that environmental factors may influence each kingdom differently. Differences between soil origins in the abundances of bacterial taxa but not fungal taxa at the phylum level support the hypothesis that members of these two kingdoms respond differently to environmental pressures. Our results show the relative abundances of AMF to be elevated in the rhizosphere samples, supporting our hypothesis of enriched symbionts in the rhizosphere. However, contrary to our hypotheses, FUNGuild assignments of indicator taxa showed that specific pathotrophic fungi were enriched in the rhizosphere. Additionally, the most abundant bacterial and fungal genera in our rhizosphere samples contain putative pathogenic taxa (*Streptomyces* and *Fusarium*). Furthermore, no significant differences for the symbiotroph trophic mode were found between soil origins, though a trend of higher relative abundances in the rhizosphere was found. Our results identify potential keystone bacteria and show that bacteria and fungi may respond differently to the selective pressures of the rhizosphere.

Across all samples, we found *Fusarium* to be the most abundant fungal genus. Our research was conducted at one of the University of Wyoming's research and extension facilities, with other potential host crops being grown in the same soils over the past several growing seasons. Having acknowledged this, previous research found a similar phenomenon [25,45], with *Fusarium* being one of the dominant genera in soils under potato. This suggests that even though our experimental plots may have been enriched due to the planting of successive susceptible crop species, the abundance of *Fusarium* spp. is likely further elevated in potato fields due to host availability [46]. While we expected to discover differences in the dominant fungal taxa between the bulk and rhizosphere soil samples, this result is not all that surprising due to the growth morphology of many fungi and the fact that these facultative pathogens are likely saprobes in the absences of a suitable host [47]. With hyphal growth, the taxa found in the rhizosphere grow into the bulk soils, resulting in the same fungal taxa being found in both soil origins.

Unlike fungi, the most common bacterial genera in rhizosphere and bulk soil samples were different, supporting our initial hypothesis of microbial assemblage differentiation as a function of soil origin. *Streptomyces* was an indicator of the rhizosphere samples, and the most abundant member of that genus was *Streptomyces scabiei*, a potato pathogen that causes potato scab [48]. While differences in the most abundant genus between soil origins supports the theory of selectivity in the rhizosphere, the abundance of *Streptomyces scabiei* in the rhizosphere samples contradicts our hypothesis of finding fewer pathogens in this soil origin. However, as this pathogen needs to infect its host through direct contact, an increase in the abundance of this pathogenic taxa and others may be expected. While *Streptomyces scabiei* is a problematic pathogen in potato production, other members of *Streptomyces* genus are thought to be plant growth promoters [49], and may provide benefits to their host plant.

In bulk soils, *RB41* was the most abundant genus of the bacterial indicator taxa. This taxon has been shown to negatively affect N assimilation, incorporating less nitrogen into its microbial biomass [50], and is found in soils with soy and corn rotation [51]. Its high abundance could be an artifact of the crop rotations in our experimental fields, which include soybeans, dry beans, and corn. Zhao et al. found a similar result when comparing continuous potato cropping systems to a corn-potato rotation [52], with *RB41* being elevated in the corn-potato annual rotation as compared to the continuous potato monoculture. Their findings suggest that the abundance of this genus is likely corn-dependent, as opposed to a lineage specifically enriched by potato.

The use of cooccurrence networks to identify important microbial nodes showed only a few fungal taxa to possess significant edges, with none having more than one significant network edge. On the other hand, the bacterial network produced 175 significant nodes, with twenty having upwards of seven significant edges. We examined the nodes with the largest number of significant edges (edges $\geq 6$), as these taxa may serve as keystone microbial taxa [53] and be critical to ecosystem structure and function, and even impede fungal pathogen establishment [54]. As may be expected, there was clear differentiation between the two soil origins with respect to the relative abundance of the taxa identified as important by our networks. In the rhizosphere samples, members of the phyla Actinobacteria and Proteobacteria showed the highest relative abundances, with *Lechevalieria* and *Streptomyces* being the most abundant genera. Members of *Lechevalieria*, specifically *Lechevalieria rhizosphaerae*, have been identified and isolated from the rhizosphere of wheat [55] and are known to produce antibiotics [56], though the full range of host associations and exact ecology of this genus still remain elusive.

Another bacterial node with multiple significant edges was a member of the genus *Massilia*. Previous research has shown *Massilia* and other members of Oxalobacteraceae to be a rhizosphere colonizers [57]. Other studies examining isolates generated specifically from potato rhizosphere showed *Massilia* to be capable of cellulose, pectin and soluble starch degradation via exoenzyme production [58]. These enzyme activities are hypothesized to increasing nutrient availability to host plants providing a fitness advantage.

Furthermore, they also showed that the assayed members of *Massilia* produced signaling molecules such as indole-3-acetic-acid (IAA), ACC deaminase, and acetoin, all of which can promote plant growth and protection. Further experimentation also showed that when soil was inoculated with a mixture of *Massilia*, *Rhizobium*, and *Chtinophaga* spp. root lesion severity on radish seedlings caused by *Rhizoctonia* decreased as compared to their control treatments. Of particular interest, the authors reported that these taxa showed no antibiotic production when cultured on media, suggesting that this consortia may control pathogenic microbes via competitive exclusion as opposed to antibiotic production [58]. As the *Massilia* genus was yet again found to be associated with the rhizosphere samples of potato, it presents itself as a potential target for an industrialized biologic amendment. However, further work is needed to accurately characterize the phenotypic characteristics of all members of the genus and understand how *Massilia* interacts with other soil organisms resulting in emergent properties [59–61].

Our results show that not only are the bulk and potato rhizosphere soil compartments different, but the differences are observable at coarse taxonomic levels for bacteria, suggesting that membership in the rhizosphere and bulk environments is selected upon by deeply conserved traits [62]. We also found that only fungal β-diversity, not bacterial, was significantly predicted by moisture content, indicating that each kingdom may respond differently to environmental ques. Our identification of putative keystone taxa provides the groundwork for future isolation and inoculation work to better understand the ecology of these microbes in relation to potato and other crops. As potato represents one of the most important staple crops around the world, understanding microbial associations represents an avenue for increasing yield via non-chemical means.

**Supplementary Materials:** The following are available online at https://www.mdpi.com/article/10.3390/applmicrobiol1020013/s1. Supplementary Table S1: Differentially abundant fungal taxa at the genus level ($p < 0.05$) as per DESeq2, and Table S2: Differentially abundant bacterial taxa at the genus level ($p < 0.01$) as per DESeq2.

**Author Contributions:** Conceptualization, G.F.C. and W.S.; methodology, G.F.C.; software, G.F.C.; validation, G.F.C.; formal analysis, G.F.C., L.T.A.v.D.; investigation, G.F.C., W.S., L.T.A.v.D.; resources, W.S., L.T.A.v.D.; data curation, G.F.C.; writing—original draft preparation, G.F.C.; writing—review and editing, G.F.C., L.T.A.v.D., W.S.; visualization, G.F.C.; supervision, L.T.A.v.D., W.S.; project administration, W.S.; funding acquisition, W.S., L.T.A.v.D. All authors have read and agreed to the published version of the manuscript.

**Funding:** This research was funded by the Microbial Ecology Collaborative, with funding from NSF grant EPS-1655726 to L.T.A.v.D., the University of Wyoming's Plant Sciences departmental graduate assistantships, and Western Potato (Alliance, NE).

**Institutional Review Board Statement:** Not applicable.

**Informed Consent Statement:** Not applicable.

**Data Availability Statement:** Raw sequence reads are available under the following NCBI SRA bioproject numbers: Bacteria: PRJNA731443 and Fungi: PRJNA733452.

**Conflicts of Interest:** The authors declare no conflict of interest.

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
