# Peer review of "An Examination of Fungal and Bacterial Assemblages in Bulk and Rhizosphere Soils under Solanum tuberosum in Southeastern Wyoming, USA"

_2673-8007, doi:10.3390/applmicrobiol1020013_

Round 1

Reviewer 1 Report

The article is well written and contains interesting data. 
Two misspelling corrections are suggested:

Row 196: istead of rhizospher, rhizosphere

Row 252: use capital letter at the beginning of the sentence. 

Author Response

Thank you for the minor edits. Your suggestions have been fixed. 

Reviewer 2 Report

  1. Than materials and methods are overly long and could be shortened considerably.
  2. The replication numbers should be made clearer to the reader.
  3. The results section could be shortened considerably.
  4. The results once shortened should be combined with the discussion.

Author Response

The materials and methods are overly long and could be shortened considerably.

Response: We believe the methods contain no superfluous aspects and as reviewer 3 indicates contains all the necessary information to repeat the experiment. We did edit a few sentences throughout to shorten slightly.

The replication numbers should be made clearer to the reader.

Response: We report the number of independent samples for each analysis. For the fungal results, “We report nine independent bulk and seven independent rhizosphere samples for our statistical analyses.” For the bacterial results “We report 8 independent bulk and 7 independent rhizosphere bacterial samples for our statistical analyses, as one sample failed to sequence.” Please see lines 176-177 and 243-244.

The results section could be shortened considerably.

We believe the results section to contain no superfluous aspects We did edit a few sentences to shorten slightly. Specifically we have condensed the third paragraph of the fungal results. See liens 190-199.

The results once shortened should be combined with the discussion.

Response: The formatting guide provided by agriculture has a separate results and discussion section. As such, we have opted to keep these sections separate. 

Reviewer 3 Report

The manuscript submitted for revision examines fungal and bacterial diversity using high throughput sequencing in soils under a potato crop grown in Southeastern Wyoming. It is an interesting subject; moreover, potato is a traditional crop with great importance worldwide. The study has the structure recommended by the journal, but the sections are not arranged properly. Please, check and re-order.

In the beginning, the abstract is well structured and gives the necessary information to the readers. The keywords are well selected.

The introduction covers all experimental parts of the manuscript, and the literature references are well selected and sufficiently detailed.

The Materials and methods section is developed in detail, understandable, and could be easily reproduced. Samples were collected from a potato field. Soil DNA was extracted, and bacterial and fungal libraries were prepared. Followed the PCR of the V4 region of the 16S rRNA gene of the bacterial genomes, the products were verified, cleaned, and quantified. The libraries were sequenced on the 122 Illumina MiSeq platform. The results were analyzed in R. Statistical analysis of differential bacterial and fungi abundance was performed using The DeSEQ2 package. Shannon diversity index and species richness were estimated through Phyloseq function estimate_richness, while the statistical differences were determined using a one-way ANOVA.

Line 157: Check the spelling.

The results are clearly presented, including fungi and bacterial PCoA, summary statistics of FUNGuild assignments, abundance, etc.

The extensive discussion section is well developed, and the main results are compared to the literature findings in the field.

The results obtained and the discussion supports the conclusions, although in my opinion, the conclusions need to be improved in terms of being more in-depth and covering the concrete results and discussion.

References need to be rewritten according to the Instructions to authors of the Agriculture. In some places, page or article numbers missing or unknown symbols are displayed (lines 440, 454, 505).

Author Response

The manuscript submitted for revision examines fungal and bacterial diversity using high throughput sequencing in soils under a potato crop grown in Southeastern Wyoming. It is an interesting subject; moreover, potato is a traditional crop with great importance worldwide. The study has the structure recommended by the journal, but the sections are not arranged properly. Please, check and re-order.

Response: Thank you for this positive assessment. Our manuscript is formatted according to the Agriculture template. We believe it the sections to be in the correct order.

In the beginning, the abstract is well structured and gives the necessary information to the readers. The keywords are well selected.

The introduction covers all experimental parts of the manuscript, and the literature references are well selected and sufficiently detailed.

The Materials and methods section is developed in detail, understandable, and could be easily reproduced. Samples were collected from a potato field. Soil DNA was extracted, and bacterial and fungal libraries were prepared. Followed the PCR of the V4 region of the 16S rRNA gene of the bacterial genomes, the products were verified, cleaned, and quantified. The libraries were sequenced on the 122 Illumina MiSeq platform. The results were analyzed in R. Statistical analysis of differential bacterial and fungi abundance was performed using The DeSEQ2 package. Shannon diversity index and species richness were estimated through Phyloseq function estimate_richness, while the statistical differences were determined using a one-way ANOVA.

Line 157: Check the spelling.

Response: DeSeq has been corrected to DESeq throughout the manuscript.

The results are clearly presented, including fungi and bacterial PCoA, summary statistics of FUNGuild assignments, abundance, etc.

The extensive discussion section is well developed, and the main results are compared to the literature findings in the field.

The results obtained and the discussion supports the conclusions, although in my opinion, the conclusions need to be improved in terms of being more in-depth and covering the concrete results and discussion.

Response: We have added additional information to the conclusion paragraph and feel that it summarizes our results and avenues of future research.

References need to be rewritten according to the Instructions to authors of the Agriculture. In some places, page or article numbers missing or unknown symbols are displayed (lines 440, 454, 505).

Response: All references have been checked and updated. The bibliography and in text references were generated using Mendeley with the Agriculture format and should be correct.